# The GDAP1 p.Glu222Lys Variant-Weak Pathogenic Effect, Cumulative Effect of Weak Sequence Variants, or Synergy of Both Factors?

**DOI:** 10.3390/genes13091546

**Published:** 2022-08-27

**Authors:** Dagmara Kabzińska, Katarzyna Chabros, Joanna Kamińska, Andrzej Kochański

**Affiliations:** 1Neuromuscular Unit, Mossakowski Medical Research Institute Polish Academy of Sciences, 02-106 Warsaw, Poland; 2Institute of Biochemistry and Biophysics Polish Academy of Sciences, 02-106 Warsaw, Poland

**Keywords:** CMT, *GDAP1* gene, the pathogenic effect of the mutation, mutation genetic burden

## Abstract

Charcot–Marie–Tooth disorders (CMT) represent a highly heterogeneous group of diseases of the peripheral nervous system in which more than 100 genes are involved. In some CMT patients, a few weak sequence variants toward other CMT genes are detected instead of one leading CMT mutation. Thus, the presence of a few variants in different CMT-associated genes raises the question concerning the pathogenic status of one of them. In this study, we aimed to analyze the pathogenic effect of c.664G>A, p.Glu222Lys variant in the *GDAP1* gene, whose mutations are known to be causative for CMT type 4A (CMT4A). Due to low penetrance and a rare occurrence limited to five patients from two Polish families affected by the CMT phenotype, there is doubt as to whether we are dealing with real pathogenic mutation. Thus, we aimed to study the pathogenic effect of the c.664G>A, p.Glu222Lys variant in its natural environment, i.e., the neuronal SH-SY5Y cell line. Additionally, we have checked the pathogenic status of p.Glu222Lys in the broader context of the whole exome. We also have analyzed the impact of *GDAP1* gene mutations on the morphology of the transfected cells. Despite the use of several tests to determine the pathogenicity of the p.Glu222Lys variant, we cannot point to one that would definitively solve the problem of pathogenicity.

## 1. Introduction

The currently available gene databases contain the still rising number of variants of unknown significance (VUS), variants of uncertain effect, or variants characterized even by contradictory results. In the era of next-generation sequencing (NGS), a leading problem seems to be establishing the pathogenic status of sequence variants rather than detecting it. The missense variants resulting from single substitutions within SNPs are an especially challenging task in terms of the pathogenic status of variants.

In 2015, nearly half of deposited sequence variants in the widely used ClinVar database were categorized as VUS. Five years later, in 2020, almost 75% of ClinVar variants were classified as VUS [1]. In other words, only one in four variants identified in the patients’ causative role of certain variants may be established.

The exponentially increasing number of variants of unknown clinical relevance has led to attempts to develop systems for the comprehensive assessment of the pathogenicity of mutations involving entire genes. Unfortunately, due to the diverse functions of proteins encoded even by the same gene, the creation of a common system for the functional assessment of the effects of mutations encounters quite serious difficulties. Different mutations of the same gene typically result in different disorders (cell viability, adhesion properties, sensitivity to toxic agents, etc.). Hence, there is a need to individualize the pathogenicity assessment of individual mutations. Some authors even propose the use of a summary assessment of the mutation effect, in which the results of various functional tests are taken into account—the integrated phenotypic score [2].

The *GDAP1* gene (8q21) coding for the ganglioside differentiation-associated protein 1 was first characterized as the gene whose expression is tightly correlated with the differentiation of the neuroblastoma cells [3]. In 2002, two independent groups identified the pathogenic mutations in the patients affected with a severe form of peripheral neuropathy with an autosomal recessive trait of inheritance also called Charcot–Marie–Tooth disease type 4A (CMT4A) [4,5,6]. In 2005, the first *GDAP1* gene mutations inherited in an autosomal dominant trait were reported [6]. Over 23 years of extensive studies of GDAP1 biology have provided a true plethora of molecular disturbances associated with *GDAP1* mutations. For many years, the hypothesis of the role of GDAP1 function of glutathione transferase was discussed. As the protein expressed in the outer mitochondrial membrane, GDAP1 was shown to be involved in the fusion/fission processes. Some authors have shown evidence of the GDAP1 role in intracellular calcium transport. For 23 years, no functional association between the neuronal differentiation process accompanied by GDAP1 and any molecular pathways have been established till now [7]. Recently, it was shown that GDAP1 interacts with cofilin 1 and β tubulin [8]. In 2014, we reported on two small Polish CMT families in which a new Glu222Lys amino acid substitution in *GDAP1* was found. Interestingly, the c.664G>A, p.Glu222Lys variant has been shown to segregate with CMT phenotype both in dominant and recessive (together with p.Leu239Phe) traits. Till now, in the literature, there was no reported, not even one CMT -affected patient harboring the heterozygous p.Leu239Phe mutation (carrier) manifesting any CMT symptoms. However, the p.Leu239Phe mutation is a recurrent pathogenic variant occurring both in the homozygous and compound heterozygous state in the CMT-affected patients [9].

In the dominant trait, Glu222Lys amino-acid substitution was associated with the extremely mild phenotype of peripheral neuropathy limited to the lower limbs beginning after the seventh decade of life [10]. In the period of 8 years after the initial publication of the first two Polish CMT (p.Glu222Lys) families, no other cases have been reported neither in the literature nor in the ClinVar database.

Due to extremely weak clinical manifestation and the lack of further CMT families, the pathogenic character of Glu222Lys amino-acid substitution could be considered at least disputable. Although the vast majority of GDAP1 mutations are inherited in recessive traits, there are some mutations that are dominant. However, the Glu222Lys amino-acid substitution is exceptional because it could be recessive as well as dominant. The double inheritance pattern occurs very rarely in inherited disorders. For such a reason, we decided to analyze the effects of Glu222Lys amino-acid substitution in a more detailed way. We could also not exclude the possibility that the pathogenic effect of the p.Glu222Lys variant depends on the ethnic or even intrafamilial context (mutation found in east European population in two families which could be related). Previously, we checked the pathogenic effect of *GDAP1* mutation, including that resulting in p.Glu222Lys in a yeast-based model in which we classified this mutation as weak/moderate [11]. The yeast-based model was a base for further studies in which we have used two independent approaches to verify the pathogenicity of p.Glu222Lys. First, we have analyzed the genomic context of the p.Glu222Lys variant by whole exome sequencing (WES) applied to two Polish CMT families. Next, we examined the effect of Glu222Lys amino-acid substitution on the *GDAP1* expression, cell viability, sensitivity to H_2_O_2_ and the differentiation process of the neuronal cell line (human neuroblastoma SH-SY5Y cell line).

## 2. Materials and Methods

### 2.1. Patients

In family A (Figure 1a), the siblings suffered from severe early onset axonal neuropathy. The genotype of them was c.715C>T, p.Leu239Phe/c.664G>A, p.Glu222Lys variants. Their 44-year-old mother with a heterozygous p.Glu222Lys variant considered herself healthy. Neurological examination in her revealed the absence of ankle jerks. In family B (Figure 1b), two patients aged 73 and 71 revealed only diminished ankle jerks. Electromyographic analysis (EMG) performed on them showed the neurogenic pattern in the muscles. Their genotype for the *GDAP1* gene was c.664G>A, p.Glu222Lys heterozygous variant. For details, see Kabzińska et al. [10].

### 2.2. Selection of the Missense GDAP1 Gene Mutations Transmitted in the Dominant Trait as the Internal Controls for the p.Glu222Lys Pathogenic Effect

Two *GDAP1* mutations, p. His123Arg and p.Ala156Gly were selected as the positive controls of the pathogenic effect of investigated p.Glu222Lys variant. The Gln218Glu substitution was also documented only in one Korean family and associated with a weak phenotype in two patients (father and daughter). We used p.Gln218Glu as a representative variant for weak mutations. The pathogenic variants resulting in His123Arg, Ala156Gly, and Gln218Glu amino-acid substitutions have been shown to segregate with CMT disease with an autosomal dominant trait of inheritance [12,13,14,15].

### 2.3. Whole Exome Sequencing

Ethics approval was obtained from the local ethic committee, as well as the informed concern from patients from families A and B. The study was conducted in accordance with the Declaration of Helsinki.

Whole exome sequencing (WES) of the proband’s DNA was performed in line with the protocol from Illumina’s TruSeq Exome Enrichment Guide. A SureSelect Human All Exon 50 Mb Kit (from Agilent Technologies, Santa Clara, CA, USA). Hybridized fragments were bound to streptavidin beads, while non-hybridized fragments were washed out. The library was then verified on an Agilent 2100 Bioanalyzer and sequenced on a HiSeq 2000 instrument (Illumina) from Intelliseq LLC (Kraków, Poland).

### 2.4. Exome Sequence Data Analysis

We have selected 154 genes (Appendix A) for the analysis calling them ‘‘CMT genes,’’—i.e., the genes involved in pure CMT phenotype and accompanying selected syndromes with hereditary neuropathies. The CMT genes were selected on the basis of the analysis of Pubmed and available databases in which peripheral neuropathy was recorded as a constant feature of the neurodegenerative process.

### 2.5. Cell Line, Media and Growth Conditions

Neuroblastoma cell line SH-SY5Y (gift from M. Szeliga MMRI PAS) was maintained in DMEM/F12 (Sigma-Aldrich, Saint Louis, MI, USA) containing 10% FBS (Thermo Fisher Scientific, Waltham, MA, USA), 100 IU/mL penicillin, and 100 μg/mL streptomycin (Thermo Fisher Scientific, Waltham, MA, USA), +1% Non-Essential Amino Acids (NEAA) (Sigma-Aldrich, Saint Louis, MI, USA) and uridine 1 mg/mL (Sigma-Aldrich, Saint Louis, MI, USA) in a humidified tissue culture incubator at 37 °C and 5% CO_2_ atmosphere.

### 2.6. Plasmids, Mutagenesis and CRISPR

*GDAP1* cDNA was subcloned from pCMV6-XL5 GDAP1 (NM_018972) Human Untagged Clone (OriGene Technologies Inc., Rockville, MD, USA) to pIRES2-AcGFP1 Vector (TAKARA Bio, Shiga, Japan) using SacI and SalI enzymes (Thermo Fisher Scientific, Waltham, MA, USA).

Mutagenesis for mutations in the *GDAP1* gene (c.652C>G p.Gln218Glu, c.664G>A p.Glu222Lys, c.368A>G p.His123Arg, c.467C>G p.Ala156Gly) was performed using Mut Express II Fast Mutagenesis Kit V2 (Vazyme Biotech Co., Ltd., Nanjing, Jiangsu, China) according to the manufacturer’s instructions and verified by sequencing.

A *GDAP1* knockout was generated using the CRISPR method with *GDAP1* sgRNA (spCas9) CRISPR Lentivector set (with three different target sgRNA: T1, T2, and T3), pLenti-U6-sgRNA-PGK-Neo from ABM (Richmond, BC, Canada). Plasmids were multiplying in TOP One *Escherichia coli* chemically competent cells then isolated with NucleoBond Xtra Midi EF kit (Macherey-Nagel, Düren, Germany). HEK293T cells (gift from Molecular Biology Unit, MMRI PAS) were seeded approximately 3 × 10^6^ cells on 10 cm Ø plates containing a complete DMEM/F12 growth medium. Cells were growing overnight to approximately 70–80% confluence. All lentiviruses used for *GDAP1* knockout (viral particles containing plasmids that express Cas9 and a guide RNA: T1, T2, T3, and T1/2/3) were generated in HEK293T cells via calcium phosphate transfection using two viral packaging plasmids ∆8.9 and VSVG (gift from Molecular Biology Unit, MMRI PAS). The viruses were collected after 24 h, and viral supernatants were filtered using a 0.45-μM filter and stored at −80 °C before SH-SY5Y cells transduction. The SH-SY5Y cells transduction was performed by adding 2 mL of viruses with different sgRNAs to 2 mL of complete DMEM/F12 growth medium on 6 cm Ø plates with ~50% confluent SH-SY5Y cells. The medium was changed 24 h after transfection and selected 48 h after transduction for Cas9 expression with 500 μg/mL G418 (Lab Empire, Rzeszów, Poland) for two weeks. The mixture of T1, T2, and T3 sgRNA viruses characterized by the most effective reduction of the GDAP1 protein level was selected for further analyses.

### 2.7. Cell Transfection

SH-SY5Y cells were transfected using the pIRES2-AcGFP1 Vector with Viromer Red (Lipocalyx, Halle, Germany) according to the manufacturer’s recommendations with the standard transfection scale. In all cases, the medium was changed 48 h after transfection. The resulting transfectant SH-SY5Y cells were cultured in an appropriate medium with 500 μg/mL G418 (Lab Empire, Rzeszów, Poland) for two weeks.

### 2.8. Protein Extracts and Western Blot Analysis

Total protein cell extracts were prepared using a cell lysis buffer (Cell Signaling Technology, Inc., Danvers, MA, USA) with 1 mM PMSF and Protease Inhibitor Cocktail 1/200 *v*/*v* (Sigma-Aldrich, Saint Louis, MI, USA). Proteins were separated by SDS-PAGE, transferred onto nitrocellulose membrane Amersham Protran (GE Healthcare Bio-Sciences AB, Uppsala, Sweden), and analyzed by Western blotting using rabbit polyclonal anti-GDAP1 (Abcam, Cambridge, MA, USA) and secondary anti-rabbit IgG horseradish peroxidase (HRP)-conjugated antibody (Sigma-Aldrich, Saint Louis, MI, USA) followed by enhanced chemiluminescence (Western Bright Sirius Advansta, San Jose, CA, USA).

### 2.9. Cells Viability

SH-SY5Y cells viability was measured using the trypan blue dye exclusion test in cells suspension by automated cell counter Countess II (Invitrogen, Camarillo, CA, USA). SH-SY5Y cells were counted and diluted to target approximately 2 × 10^6^ cells/mL concentration. The 10 µL of 0.4% trypan blue (Sigma-Aldrich, Saint Louis, MI, USA) and 10 µL of cells suspension were mixed and incubated ~3 min at room temperature. After incubation, cells were immediately applied on disposable slides, and the percentages of viable cells were calculated automatically.

### 2.10. Sensitivity of Cells to H_2_O_2_

The amount of 4 × 10^5^ cells was suspended in 200 µL of PBS (Sigma-Aldrich, Saint Louis, MI, USA) or PBS with different amounts of 30% H_2_O_2_ to concentrations: 100, 200, 300, 400, 500, and 600 mM, respectively. Cells were incubated 10 min in a humidified tissue culture incubator at 37 °C and 5% CO_2_ atmosphere, spin down 3 min at 300 g, suspended in 100 µL of PBS, and tested for viability with the trypan blue dye.

### 2.11. Live Cell Imaging

SH-SY5Y cells were plated on 6-well plates uncoated (plastic) in the density of 0.5 × 10^5^ and incubated in regular DMEM media containing 10% FBS at 37 °C, 5% CO_2_ for 168 h (7 days). Pictures were taken at 20× using Zeiss Axio Imager brightfield microscope and ZEN 3.1 (blue edition) software at 24 (Day 1), 48 (Day 2), 72 (Day 3), 120 (Day 5), and 168 (Day 7) hours after cells seeding. Scale bar = 100 μm.

### 2.12. Statistical Analysis

Statistical analyses were performed using GraphPad Prism Software (San Diego, CA, USA) (https://www.graphpad.com/scientific-software/prism/, Accessed on 10 May 2022) using one-way ANOVA and Dunnett correction: * *p* < 0.05, ** *p* < 0.01, *** *p* < 0.001, **** *p* < 0.001. Data are presented as means ± standard deviations.

## 3. Results

### 3.1. Patients with GDAP1 Causative Mutations Have Additional Weak CMT Sequence Variants

Among *GDAP1* variants are some that are poorly characterized. These include the p.Glu222Lys and p.Gln218Glu variants. We noted that, so far, there are no other reports of the p.Glu222Lys variant in the literature besides the report by Kabzińska et al. [10]. Similarly, the other analyzed by us variants, i.e., p.Gln218Glu, was reported only in one Korean family in 2008 [12]. More detailed characteristics of this Korean family were reported in 2021, supporting its pathogenic character [14]. Similarly to p.Gln218Glu, we would like to characterize the p.Glu222Lys variant more deeply, which was found only in two Polish families to the best of our knowledge.

While, in family A, we observe a severe clinical course of the disease in siblings of a similar age, but with a difference in the clinical picture, in family B, we observe a very similar and unusually mild clinical phenotype in the siblings. Interestingly, in family A, the disease is inherited both in a recessive and dominant manner, whereas in family B as a dominant trait [10]. The mildness of the phenotype, the presence of a few families that do not allow tracking the segregation of the variant with the phenotype, the marked inter-familiar clinical variability, and finally, the dual mode of inheritance does not exclude the possibility of the random coexistence of p.Glu222Lys variant with a CMT phenotype. Thus, we asked the questions of whether this mutation is really a pathogenic one and if the symptoms are the result of genetic variability, at least in other CMT genes. Therefore, to test this hypothesis, the WES studies were conducted on six members of families A and B (Figure 1a,b).

The analysis of WES revealed that the highest number (5) of additional variants in ‘‘CMT genes’’ were found in the exome of the most seriously affected patient III:2 from the family A. In the much milder affected brother III:3, only two additional variants were found in the ‘‘CMT genes’’ (Figure 1a). The mother of these siblings II:4 has two additional variants with a likely benign pathogenic status. The healthy father II:3 has three additional variants of benign, likely benign status or not reported to the databases.

In the case of the *DST* c.8626A>T variant, the pathogenicity status is unknown because it has not been reported to the databases up until now. Similarly, the databases have not yet recorded the *PRNP* c.654C>T variant. In the case of family B, the genetic load seems to be lower/low, and both patients III:9 and III:11 have the same two additional sequence variants in *DST* c.8626A>T and *PEX1* c.149G>A genes. The difference in the genetic load between the studied two families is interesting.

Due to the identity of mutations in patients III:9 and III:11 in family B, it can be presumed that they were inherited from one of the parents.

As presented in this study of two families, we see that the number of CMT variants correlated with the clinical picture. Similarly, in the larger group of CMT families, we observed a positive correlation between sequence variants and the clinical picture of CMT. All the above-presented data strongly support the hypothesis that genetic background could have an additional impact on the clinical presentation of CMT.

### 3.2. The Level of GDAP1 Glu222Lys Protein in the SH-SY5Y Cell Line Is Reduced

We expected to answer the question of whether p.Glu222Lys is a leading mutation after WES, but our analysis identified mutations of unknown status or completely unknown. This does not allow us to answer the question if the p.Glu222Lys variant is a leading mutation. Is p.Glu222Lys one of many variants in CMT genes whose total effect we observe in the presented families? To more accurately answer these questions, we made a model to assess the pathogenicity of the p.Glu222Lys variant. The model is based on SH-SY5Y cells transfected with the empty vector or vector bearing cDNA of wild-type *GDAP1* and c.664G>A, p.Glu222Lys and control variants with established pathogenicity c.652C>G, p.Gln218Glu; c.368A>G, p.His123Arg; and c.467C>G, p.Ala156Gly. This is due to the fact that a decrease in the level of GDAP1 has already been observed in the fibroblasts taken from the patients with the p.Leu239Phe/p.Arg273Gly and *GDAP1* c.579+1G>A pathogenic variants [16]. The first step of our study was the analysis of the level of GDAP1 protein variants produced in SH-SY5Y cells transfected with cDNA of different *GDAP1* alleles. Additionally, the knockout *GDAP1* mutant was used as a control of GDAP1 protein level reduction. The level of Glu222Lys protein in the SH-SY5Y cells, together with levels of three other GDAP1 protein variants, as a control, was analyzed by Western blot. The cells transfected with cDNA of the wild-type *GDAP1* show the highest level of GDAP1. In the case of three pathogenic *GDAP1* variants, i.e., p.Gln218Glu, p.His123Arg, and p.Ala156Gly, respectively, the level of GDAP1 was reduced. A highly reduced level of GDAP1 was observed for the analyzed p.Glu222Lys variant.

### 3.3. The Reduced Cell Viability and Increased Sensitivity to H_2_O_2_ of the SH-SY5Y Cell Line Transfected with Different GDAP1 Alleles

In 2012, Noack et al. showed that *GDAP1* overexpression increases the cellular level of GSH in the HT22 cells and stabilizes the mitochondrial membrane potential by decreasing the production of reactive oxygen species (ROS) [16]. In light of these findings, the pathogenic *GDAP1* gene mutations by reduction of GDAP1 level (Figure 2) could result in high levels of damaging ROS and, consequently, reduce cell viability. Moreover, the *GDAP1* mutants may be more vulnerable to the ROS overproduction resulting from the exposition of the cells to H_2_O_2_-the well-established ROS-producing agent.

We checked this hypothesis for four *GDAP1* gene variants, i.e., p.Gln218Glu, p.Glu222Lys, p.His123Arg and p.Ala156Gly. Interestingly, the overexpression of theoretically protective wild-type *GDAP1* resulted in a slight reduction of SH-SY5Y cell viability (Figure 3). The expression of two variants, i.e., p.His123Arg and p.Ala156Gly resulted in the reduction of viability of the SH-SY5Y cells. The Gln218Glu and Glu222Lys amino-acid substitutions resulted in a milder reduction of cell viability compared to GDAP1 overproduction, which was of no statistical significance. In the analysis of sensitivity to the H_2_O_2_ of the transfected SH-SY5Y cells, the strongest effect was observed again for p.His123Arg and p.Ala156Gly variants. The expression of p.Glu222Lys resulted in a higher sensitivity to H_2_O_2_ exposition compared to the wild-type control SH-SY5Y cells. The cells expressing the p.Gly218Glu variant were almost not sensitive to the H_2_O_2_. As could be expected, the knockout *GDAP1* mutant was the most sensitive to H_2_O_2_.

To conclude, the overexpression of the *GDAP1* in SH-SY5Y cells leads to the increased sensitivity of the cells to H_2_O_2_. All the analyzed *GDAP1* missense variants resulted in both reduced viability and increased sensitivity to H_2_O_2_. The experiment showed no correlation between the reduction of GDAP1 levels and sensitivity to H_2_O_2_ and cell viability.

### 3.4. Altered Morphology of Cells Expressing GDAP1 p.Glu222Lys Mutant Allele

Given the results from studying cellular dysfunction were not completely informative, we decided to analyze more phenotypes and look closer at the cell’s morphology. Based on the first study concerning GDAP1 protein showing that GDAP1 plays a role in the process of differentiation of the neuronal cells [3], we expected that various levels of different GDAP1 protein variants would also affect cell morphology. Undifferentiated SH-SY5Y cells tend to grow in clusters and may form clumps, while differentiated SH-SY5Y neurons have a more pyramidal-shaped cell body and demonstrate extensive and elongated neuritic projections. Differentiated SH-SY5Y cells consist of two morphologically distinct types: ‘‘S’’ and ‘‘N.’’ The ‘‘N’’ type is more neuronal-like with a pyramidal shape and long projections, while, in contrast, ‘‘S’’ type epithelial-like cells have no neurites like projections [17]. We have checked the sprouting process of the SH-SY5Y cells transfected with different *GDAP1* alleles and a knockout of *GDAP1*. It is noteworthy for the whole period of cell culture that the cells expressing p.Glu222Lys did not sprout efficiently (Figure 4) on the 1,2,3,5, and 7th days of observation (Appendix A). This observation seems to be in line with the previously observed significant reduction of p.Glu222Lys expression. The remaining analyzed *GDAP1* mutants did not show any obvious sprouting abnormalities. To conclude, the analysis of morphology provided the most obvious and clear interpretable results regarding the pathogenic status of p.Glu222Lys variant.

## 4. Discussion

The still existing difficulties in the assessment of the pathogenic status of the rare sequence variants are especially visible for the missense mutations segregating with mildly expressed clinical phenotype. This issue is becoming increasingly difficult with different mutations in several genes resulting in a similar clinical picture. In these patients, NGS analysis can reveal numerous equally important sequence variants presenting a benign phenotype. How to establish whether the certain phenotype results from a cumulative effect of a series of equally-weighted mutations or is caused by common action of a single leading mutation and weak sequence variants remains to be answered. In this study, we noted that genetic background influences the pathogenicity of the *GDAP1* p.Glu222Lys variant. Subsequent experiments to determine the pathogenicity of the p.Glu222Lys variant like determining the viability of cells and their sensitivity to H_2_O_2,_ did not provide conclusive results. However, the following experiments showed morphological changes in the cells, which were accompanied by a significant reduction in the GDAP1 protein level. It seems that, in the case of the studied p.Glu222Lys variant, the analysis of GDAP1 protein levels together with morphological evaluation provides valuable arguments for the pathogenicity at least of this mutation.

Here, we studied whether p.Glu222Lys variant is a leading mutation necessary for the CMT phenotype or whether it represents only one of the sequence variants, which interact with other weak sequence variants detected in other CMT genes. Indeed, in this study, we have shown that the number of weak CMT sequence variants correlates with the severity of the CMT phenotype at least in the families A and B. In the former studies the effect of additional weak variants in CMT disorders has been also clearly shown [18]. In particular, the cumulative effect of *GDAP1* and *MFN2* gene sequence variants was established. [19,20,21,22]. Such a genetic interaction is observed when proteins encoded by mutated genes participate in the same process. This is a case for GDAP1 and MFN2 proteins, which are implicated in mitochondria fusion and fission processes. Some authors go even further and suggest that, at least in some patients, the CMT phenotype may result from the coexistence of numerous not functionally related sequence variants transmitted in polygenic traits [23]. To conclude in our study, we have shown that the final effect of p.Glu222Lys variant depends also on the genetic burden of the mutations present in other CMT genes.

Not only do the various sequence variants contribute to the phenotype, but genetic interaction between two alleles of *GDAP1* is important for the genetic load. Even the variant p.Glu222Lys seems to be weak in recent studies studying the interaction of the amino-acid GDAP1 residues, and the authors showed that the Glu222Lys residue is sandwiched between three Arg side chains (Arg120, Arg225, Arg226) and Tyr124, and, additionally, this residue has van der Waals contacts to Leu239, the amino acid residue most frequently substituted in CMT4A [24]. This implies the importance of Glu222 amino acid residue in maintaining the proper structure of GDAP1 protein, and importantly its substitution can affect Leu239Phe residue. Thus, in patients from family A, the existence of compound heterozygote can mimic the homozygous mutations resulting in amino-acid substitutions of Leu239Phe. Indeed, the observed clinical phenotype in family A (III:2 and III:3) is similar to the patients harboring homozygous mutations causing substitutions of Leu239 residue. We presented only some aspects of genetic burden, which in our study is limited to CMT genes. The genetic burden may also involve other genes and variations of regulatory genome sequences. In more general terms, the interactions (synergy) between various alterations within CMT genes may be interpreted in light of the human disease network (HDN) conception. In other words, the genes being expressed together in specific tissues contribute to a final phenotype. Thus, the sequence variants we found toward CMT genes cooperate in a common system (peripheral nerve). The combined effect of weak variants located within CMT genes may impair the buffering capability of the system, in which various mutational events are coincident in a personal genome [18,25].

Since the studies of genetic load did not provide a clear explanation of the pathogenic nature of p.Glu222Lys variant, we have investigated the effect of the Glu222Lys amino acid substitution in the more natural *milieu* of the human neuronal cells i.e., in the neuroblastoma SH-SY5Y cell line. We have shown that expression of some of the *GDAP1* mutants analyzed by us resulted in reduced viability of the cells. The SH-SY5Y transfected cells were sensitive to the exposure to the H_2_O_2_ with the highest sensitivity displayed by cells having a knockout of the *GDAP1* gene. These findings are in line with observations made previously by Noack et al. for other *GDAP1* gene mutations [16]. Surprisingly, we did not observe the protective effect of the *GDAP1* gene overexpression on survival in the presence of H_2_O_2_. However, we used the different cell lines to model the effect of mutations (HT22; the immortalized mouse hippocampal cell line, versus SH-SY5Y; the human neuroblastoma cell line). The reduced viability upon H_2_O_2_ treatment has to be related to the level of GDAP1 protein variants. In contrast, we see that p.Glu222Lys and p.Gln218Glu, despite very low expression, result in a weaker sensitivity than *GDAP1* overexpression, which results in high levels of GDAP1 protein. Moreover, the effect of p.Glu222Lys variant observed in the cell-line model of CMT4A disease does not correspond with the clinical severity of CMT4A. Within four dominantly inherited *GDAP1* analyzed variants, the reduction of p.Glu222Lys protein level is the biggest one. If *GDAP1* gene mutation will be acting in gene dosage fashion, the clinical phenotype for Glu222Lys amino-acid substitution should be the strongest one, but it is not. However, in general, the gene dosage effects are observed in CMT1A disease caused by duplication of the *PMP22* gene and in CMT2S caused by the mutations in the *IGHMBP2* gene [26,27].

Due to the fact that, in the first study devoted to the GDAP1 protein, the authors showed the correlation between neurite -like differentiation and sprouting of the Neuro2a cells and expression of *GDAP1* gene [3]. Following this lead, we decided to check whether the p.Glu222Lys variant resulting in a severely reduced level of GDAP1 protein will evoke the abnormalities in the neural-like cell differentiation and sprouting of the SH-SY5Y cells. This lead turned out to be the right one. The p.Glu222Lys transfected SH-SY5Y cells did not differentiate and did not show any signs of sprouting. This effect was to be observed only for p.Glu222Lys variant. Interestingly, the relationship between *GDAP1* expression and neural sprouting may be generalized because it is not limited to the primarily examined Neuro2a cells. It is of note that the strong effect of p.Glu222Lys variant observed for the morphology of SH-SY5Y cells could not be observed at the clinical level. In the light of these findings, everything shows that the process of neuronal differentiation is not a critical issue in the pathogenesis of CMT4 disease and that the other more pathogenic role of GDAP1 remains to be discovered. It is worth noting that a severe reduction in GDAP1 protein levels, as in the case of Glu222Lys amino-acid substitution, is accompanied by morphological changes. In recent studies by Miressi et al., the mutation resulting in p.Ser194* protein truncation was shown to be associated with a low level of protein, probably due to nonsense-mediated RNA decay. This also caused disturbances in the differentiation of motor neurons from hiPSCs [28].

Nevertheless, the morphological phenotype associated with the p.Glu222Lys variant may serve as the measure of the pathogenic effect of this variant but seems to be not serviceable in any attempts trying to establish the clinical prognosis of CMT-GDAP1 types. Interestingly, Glu222Lys substitution has a similar effect as mentioned above GDAP1 protein truncation.

## 5. Conclusions

Our studies show that, even for a single gene, no general functional test of mutation pathogenicity can be established. We see that the mechanism of action of the p.His123Arg and p.Ala156Gly mutations are quite different from the mechanism for the p.Gln218Glu or p.Gly222Lys variants. In the case of p.His123Arg and p.Ala156Gly, we see a significant reduction of cell viability, which is not present for p.Gln218Glu and p.Glu222Lys. On the contrary, p.His123Asp and p.Ala156Gly do not affect cell sprouting, which does not exclude their obvious and proved pathogenicity, verified by other functional assays. We have shown that assessing the pathogenicity of mutations requires a highly individualized approach.

We assume that a proper test of sequence variant’s pathogenicity should be selected very carefully, taking into account the most impaired function or cellular process disturbed by mutation. The pathogenic status of certain sequence variants should be assessed in the cellular *milieu*, which closely resembles the natural protein environment. For some weak sequence variants, even the combination of various pathogenicity approaches may not be sufficient to establish the causative role of mutation. Thus, especially in cases of weak variants, reporting of the clinical picture of unrelated patients harboring the same variant seems to be crucial in the process of assessing of causative role of the certain mutation.

There is an urgent need to add the results of functional tests concerning the pathogenic effect of mutations to existing databases and correlate them with available clinical and genealogical data.

## Figures and Tables

**Figure 1 genes-13-01546-f001:**
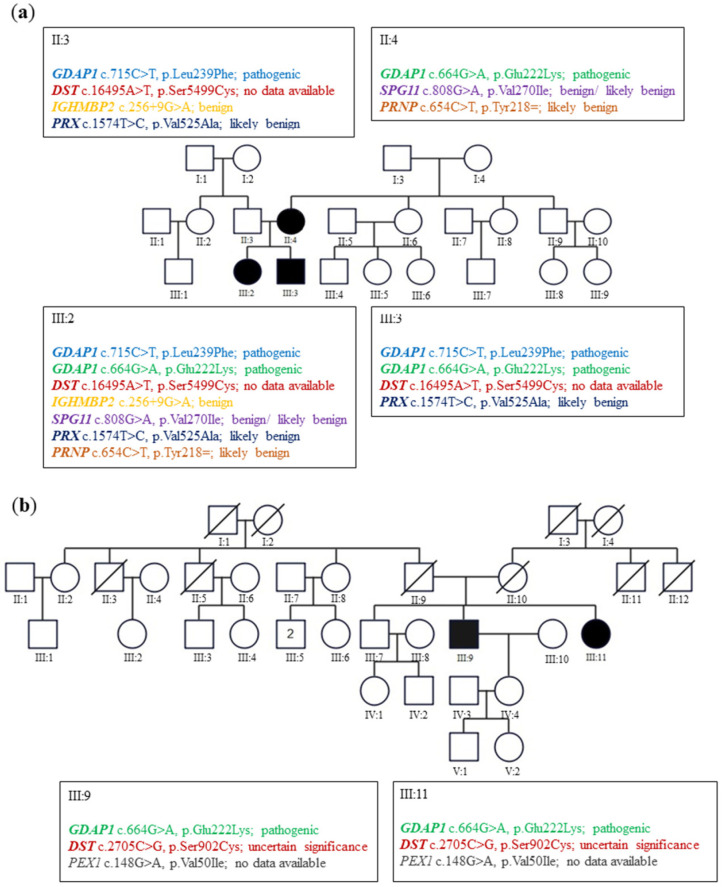
Pedigrees of the studied families A and B. (**a**) In family A, the p.Glu222Lys variant was found in the II:4 in a heterozygote and III:2 and III:3 in the compound heterozygosity with p.Leu239Phe. (**b**) In family B, the heterozygous p.Glu222Lys variant was found in III:9 and III:11. In both families, some additional sequence variants in CMT genes: *DST* (encoding Dystonin), *IGHMBP2* (Immunoglobulin Mu DNA Binding Protein 2), *PRX* (Periaxin), *SPG11* (SPG11 Vesicle Trafficking Associated, Spatacsin), *PRNP* (Prion protein) and *PEX1* (Peroxisomal Biogenesis Factor 1) were found with the use of WES. Circles mark women, squares mark men, the crossed-out symbol means a deceased person, black symbols mean sick people, and a number in a square represents the number of healthy people of a given sex. Roman numerals indicate generation, and Arabic numerals indicate successive persons in a given generation.

**Figure 2 genes-13-01546-f002:**
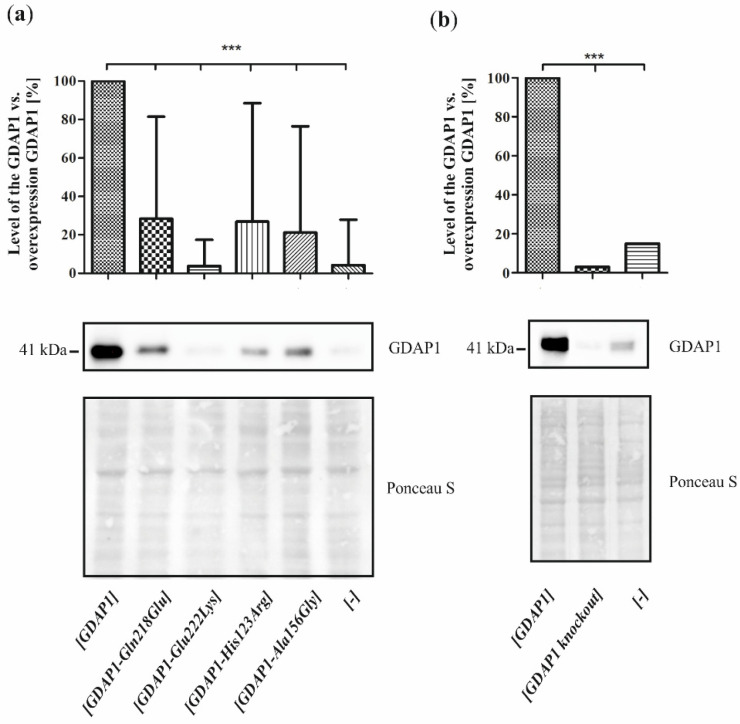
Different levels of GDAP1 protein in SH-SY5Y cells transfected with cDNAs of different *GDAP1* variants. Western blot analysis of GDAP1 level in protein extracts from transfected SH-SY5Y cells with anti-GDAP1 antibody. Ponceau S staining of the blot shown as a loading control. Densitometric analysis of the GDAP1 protein level in SH-SY5Y cell line transfected to express indicated *GDAP1* alleles. (**a**) the different variants of the *GDAP1* gene; (**b**) the knockout of the *GDAP1* gene. The relative level of GDAP1 mutant proteins to wild-type GDAP1 in SH-SY5Y cells. The results are means from three repetitions ± standard deviations; Statistical significance was determined using one-way ANOVA and Dunnett correction with *** *p* < 0.001 vs. GDAP1 overexpression SH-SY5Y cells.

**Figure 3 genes-13-01546-f003:**
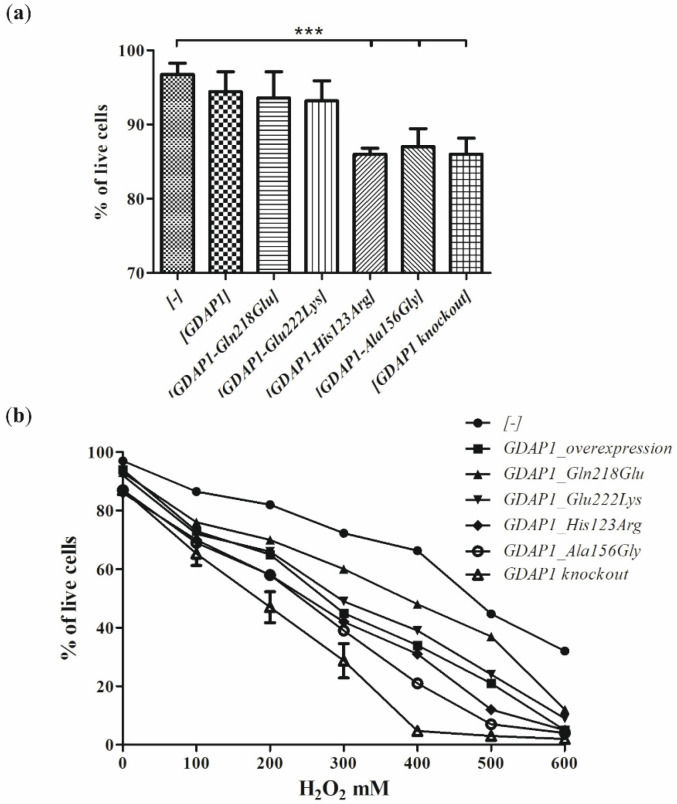
The viability and sensitivity to H_2_O_2_ of the SH-SY5Y cell line expressing the *GDAP1* mutant alleles. Trypan blue dye exclusion test of *GDAP1* (*GDAP1_overexpression*) or *GDAP1* missense variants expressing cells and *GDAP1* knockout cells compared with the control cells with empty vector [-] (**a**) from standard growth conditions (**b**) after treatment with H_2_O_2_. The results are means from three repetitions ± standard deviations; Statistical significance was determined using one-way ANOVA and Dunnett correction with *** *p* < 0.001 vs. empty vector [-].

**Figure 4 genes-13-01546-f004:**
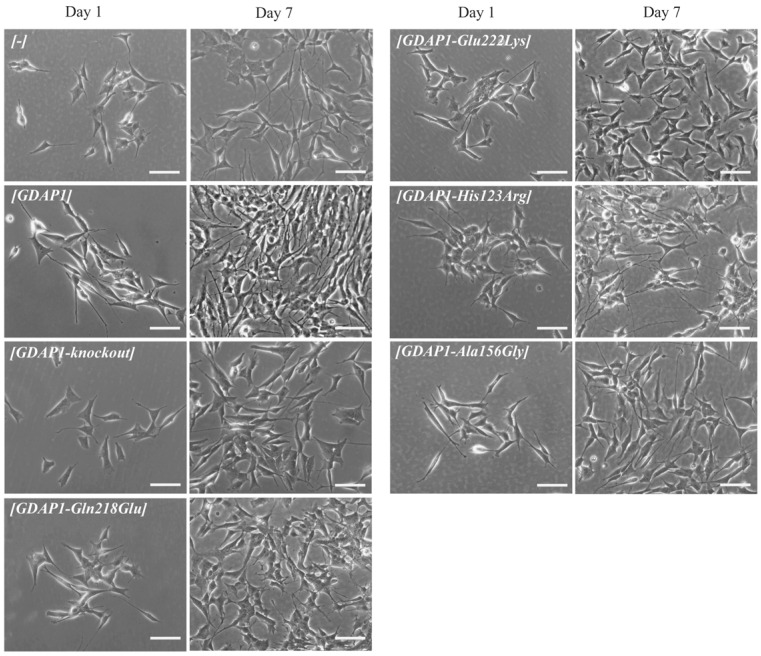
The expression of p.Glu222Lys results in a marked reduction of neurites-like projections formation. The SH-SY5Y cells with a knockout of *GDAP1* or transfected with indicated *GDAP1* alleles were visualized under light microscopy Zeiss Axio Imager at 24 (Day 1) and 168 (Day 7) hours after cells seeding. Scale bar = 100 μm.

## Data Availability

The data provided in this study are available on request from D.K. These data are not available due to privacy.

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
