# Peer review of "The GDAP1 p.Glu222Lys Variant-Weak Pathogenic Effect, Cumulative Effect of Weak Sequence Variants, or Synergy of Both Factors?"

_genes, 2022, doi:10.3390/genes13091546_

Round 1

Reviewer 1 Report

The authors analyzed the weak pathogenic effect of p.E222K as an underlying cause of CMT4A. The effect of p.E222K was examined by several available methods, such as GDAP1 expression, cell viability, sensitivity to H2O2 and the differentiation process of the neuronal cell line (SH-SY5Y), but still they obtained clear answer. However, this study is regarded to be of a value in that it attempted to address deeply the pathogenicity of a weak variant with dominant and recessive genetic patterns.

 In Figure 1a, two affected siblings’ father (II-3) was unaffected, but GDAP1 p.Leu239Phe was indicated by pathogenic. It seems to be discrepancy. I would like to recommend to correct the p.Leu239Phe into likely benign (because it was not segregated with affected individuals, although it may contribute to the CMT phenotype in the siblings with compound heterozygous mutations).

 The manuscript is somewhat confusing in description of gene and protein names and nomenclature of mutations. Please keep that gene names with italic, protein names with normal letters, Glu222Lys with normal p.Glu222Lys.

 Minor points

It is recommended to provide web addresses or cite references for many websites or programs, such as BWA-MEM software, GATK, GnomAD, MITOMAP, and COSMIC.

Please check reference format.

[Page 1, tile] Gly222Lys should be changed into p.Glu222Lys.

[Page 1, line 18] CMT type 4A disease > CMT type 4A.

[Page 1, line 24] Glu 222Lys > p.Glu222Lys.

[Page 3, line 96] For EMG, provide full words first.

[Page 3, line 107] Whole Exome Sequencing > Whole exome sequencing

[Page 3, line 113, 115; Page 5, line 193] For Agilent Technologies, Illumina, and Invitrogen, provide city and country names.

[Page 3, line 125] 200bp > 200 bp

Author Response

We highly appreciate the efforts and all the remarks of the Reviewer, which are crucial to improving the quality of our manuscript.

The status of the p.Leu239Phe mutation in the GDAP1 gene has been described in detail in the text. This is a clearly pathogenic and recurrent mutation transmitted only in a recessive fashion both in the homozygous and heterozygous state. To date in the literature no carrier of the heterozygous p.Leu239Ohe mutation harboring any CMT clinical signs has been reported [KabziÅ„ska D, Strugalska-Cynowska H, Kostera-Pruszczyk A, Ryniewicz B, Posmyk R, Midro A, Seeman P, Báranková L, ZimoÅ„ M, Baets J, Timmerman V, Guergueltcheva V, Tournev I, Sarafov S, De Jonghe P, Jordanova A, Hausmanowa-Petrusewicz I, KochaÅ„ski A.Neurogenetics. 2010 Jul;11(3):357-66 ].

We decided to exclude from the main text the details concerning WES procedure since WES was performed in the commercial and standard procedure carried out by Intelliseq Kraków, Poland.

The conclusion section has been improved to show more general aspects of our study.

The gene and protein names and mutation nomenclature have been unified and improved. We also have addressed all the minor points presented in the review.

Reviewer 2 Report

I would like to thank the Editor for the invitation to review the original manuscript entitled "The GDAP1-Gly222Lys mutation- weak pathogenic effect, cumulative effect of weak sequence variants or synergy of both factors?". 

1. My recommendation to the authors is to change the use of the term "mutation" along the text. There are several current recommendations brought by ACMG for the new designation of "variants". 

2. An important topic for discussion about this research is the potential synergy of both variants present in family A (III.2 and III.3). It would be interesting for the manuscript if the authors include a brief discussion about such mechanisms of synergy involving a monogenic basis or polygenic mechanisms (neuroprotection and/or neurodegeneration). 

3. In some sentences, abbreviation can be more properly used. For example, when describing CMT type 4A, there is no need to include the word "disease" after 4A (such as in the Abstract). 

4. The manuscript can even improve its quality by a detailed review of language aspects by a native English speaker. 

5. When describing "Gly222Lys", my suggestion is to use "p.Gly222Lys" or G222K. 

Author Response

We highly appreciate the efforts and all the remarks of the Reviewer, which are crucial to improving the quality of our manuscript.

  1. We have changed the use of the term mutation along the text using ACMG recommendations.
  2. We have added in the section of discussion a brief discussion concerning possible mechanisms of the synergy of action of weak sequence variants in the light of conceptions of the human disease network [Kwang-Il Goh, et al. PNAS 2007; 104(21): 8685-8690] and a dynamic network approach for the study of human phenotypes [Hidalgo CA, Blumm N, et al. PLoS Computational Biology 2009; 5(4) 1-11].
  3. We tried to improve the language aspect of the manuscript
  4. The nomenclature of proteins, genes, and diseases has also been improved.